# Pregnant Women’s Exposure to Household Air Pollution in Rural Bangladesh: A Feasibility Study for Poriborton: The CHANge Trial

**DOI:** 10.3390/ijerph19010482

**Published:** 2022-01-02

**Authors:** Jonathan Thornburg, Sajia Islam, Sk Masum Billah, Brianna Chan, Michelle McCombs, Maggie Abbott, Ashraful Alam, Camille Raynes-Greenow

**Affiliations:** 1RTI International, Technology-Advancement-Commercialization, 3040 Cornwallis Road, Research Triangle Park, NC 27707, USA; bchan.contractor@rti.org (B.C.); mmccombs@rti.org (M.M.); meabbott@ncsu.edu (M.A.); 2Maternal and Child Health Division, International Centre for Diarrheal Disease Research, Bangladesh (icddr,b), Mohakhali, Dhaka 1212, Bangladesh; sajiaislam@icddrb.org (S.I.); billah@icddrb.org (S.M.B.); 3Sydney School of Public Health, The University of Sydney, Edward Ford Building (A27), Camperdown, NSW 2006, Australia; Neeloy.alam@sydney.edu.au (A.A.); camille.raynes-greenow@sydney.edu.au (C.R.-G.)

**Keywords:** household air pollution, personal exposure, biomass, liquified petroleum gas, pregnancy

## Abstract

The use of liquefied petroleum gas (LPG) for cooking is a strategy to reduce household air pollution (HAP) exposure and improve health. We conducted this feasibility study to evaluate personal exposure measurement methods to representatively assess reductions in HAP exposure. We enrolled 30 pregnant women to wear a MicroPEM for 24 h to assess their HAP exposure when cooking with a traditional stove (baseline) and with an LPG stove (intervention). The women wore the MicroPEM an average of 77% and 69% of the time during the baseline and intervention phases, respectively. Mean gravimetric PM_2.5_ mass and black carbon concentrations were comparable during baseline and intervention. Temporal analysis of the MicroPEM nephelometer data identified high PM_2.5_ concentrations in the afternoon, late evening, and overnight during the intervention phase. Likely seasonal sources present during the intervention phase were emissions from brick kiln and rice parboiling facilities, and evening kerosene lamp and mosquito coil use. Mean background adjusted PM_2.5_ concentrations during cooking were lower during intervention at 71 μg/m^3^, versus 105 μg/m^3^ during baseline. Representative real-time personal PM_2.5_ concentration measurements supplemented with ambient PM_2.5_ measures and surveys will be a valuable tool to disentangle external sources of PM_2.5_, other indoor HAP sources, and fuel-sparing behaviors when assessing the HAP reduction due to intervention with LPG stoves.

## 1. Introduction

Globally, household air pollution (HAP), which includes hazardous substances such as carbon monoxide (CO), particulate matter (PM), and other environmental toxins, is the third leading health risk for mortality and the most important environmental health risk [1,2,3]. The global mortality attributable to HAP is 4.3 million deaths or 7.7% of all deaths from acute respiratory infection, ischemic heart disease, and lung cancer [1]. Approximately 3 billion people depend on biomass fuels (e.g., wood, dung, crop waste) for cooking, heating, and lighting [4,5]. The poor-quality fuels, inefficient stoves with highly polluting combustion processes, and poorly functioning chimneys for good indoor ventilation, cause chronic exposure to elevated HAP concentrations. HAP contributes to a significant health burden on various populations, namely, low- and middle-income countries (LMICS) and women of reproductive age [5,6]. According to the World Health Organization (WHO), women and children bear the greatest burden of this exposure as a particularly vulnerable group [7,8]. In South Asia, HAP exposure is the number one health risk for childhood communicable diseases, rising to prominence over poor water and sanitation and childhood micronutrient deficiencies [1].

The adverse outcomes of HAP on maternal and child health are a concerning topic regarding public health. Pregnant women and their fetuses are especially prone to adverse impacts of HAP regarding perinatal complications leading to stillbirth and neonatal mortality. The toxic particulate matter and gases can cross the placenta and reduce oxygen delivery to the fetus, hindering human development [9,10]. One study that assessed the impact of exposure to cooking fuels on stillbirths concluded there was an increased risk (absolute risk reduction (aRR) 1.44, 95% confidence interval (CI) 1.3–1.61) of perinatal mortality among households using polluting fuels [11]. Meta-analyses reported pooled estimates for stillbirth odds ratio (OR) were 1.51 (95% CI: 1.23, 1.85) and OR 1.51 (95% CI: 1.23, 1.85) [12,13]. The included studies, however, were retrospective with potential for selection bias and did not account for important risk factors [10,14,15,16].

Low birth weight, an important indicator for morbidity and mortality in infancy, has been measured in various studies to assess the impact of biomass fuel [17]. A recent study in Sri Lanka found that high exposure to HAP resulted in an increased risk of low birth weight compared to a lower exposure category, which only used clean energy as cooking fuel (aOR 3.23 (95% CI 1.17–8.89)) [18]. However, many of these studies were cross-sectional and used proxy measures instead of direct measurement of HAP exposure.

The HAP-related health outcomes associated with traditional cooking practices have encouraged adoption of alternative fuel to decrease HAP exposure. Further, ventilation, stove condition, time spent cooking, and other factors must also be studied to gain a comprehensive understanding of opportunities to reduce HAP exposure [19]. Liquid petroleum gas (LPG), a mixture of propane and butane, burns efficiently and therefore emits less pollution than a traditional biomass stove [20]. Support for improving accessibility to LPG stoves has increased in part due to improved short-term health outcomes [18,21,22,23].

It is likely that an LPG stove can help lower indoor PM_2.5_ levels to the WHO target of 10 μg/m^3^, which is highly favorable compared to other speculated interventions with improved biomass cookstoves [22,24]. Two studies have estimated a ≈33% reduction in personal exposure to PM_2.5_ in pregnant women associated with using LPG fuels instead of biomass fuels [25,26]. Likewise, studies have shown that CO concentrations decrease with the use of LPG stoves [27]. This is important because CO, which has a particularly strong affinity for hemoglobin, is a primary source of compromised oxygenation of the fetus [19]. Chronic CO exposure is also specifically linked to other outcomes, including asthma, cardiovascular disease, and neurological development [28]. Despite the benefits of LPG stoves, some studies show that overall PM_2.5_ exposure levels are still 18 times higher than the WHO standard in areas that implement LPG usage, which leads us to speculate other possible contributors to exposure [29].

The availability of LPG fuel in low-resource settings such as Bangladesh is currently limited to urban populations, whereas rural populations rely almost universally on traditional biomass fuels. Important social implications must be considered before deciding whether the widespread adoption of LPG stoves is the most viable and sustainable option for improving air quality. The impact of a clean cooking intervention on HAP exposure reduction and associated improved health outcomes depends on behavior change that leads to sustained LPG stove adoption [30,31,32]. Cultural cooking practices, willingness to pay, fuel accessibility, economic factors associated with fuel costs and stove maintenance, and inadequate estimates of HAP reduction can adversely impact any clean cooking intervention [33,34,35,36,37,38].

This study aims to provide context on how we can directly target rural, pregnant women’s HAP exposure from traditional stove use to improve perinatal and neonatal outcomes. In preparation for a prospective community-based randomized controlled clean cooking trial of perinatal and neonatal mortality and morbidity in rural Bangladesh, we conducted a feasibility study. We aimed to assess the barriers and facilitators for conducting the clean cooking trial with pregnant women, including an evaluation of methods to assess reductions in HAP exposure (Poriborton: The CHANge Trial, ACTRN12618001214224). Our objectives were to assess the data quality and information gained from personal exposure monitoring, characterize the women’s exposure distributions pre- and post-intervention, and quantify the reduction in HAP exposure that resulted from the clean cooking intervention.

## 2. Materials and Methods

The Poriborton: The CHANge Trial feasibility study is described in Raynes-Greenow (2020) [30]. To briefly summarize, pregnant women’s exposure to HAP before and during LPG stove distribution were measured. A subset of 30 women with gestational age between 12 to 20 weeks and not using LPG for cooking enrolled in the feasibility trial were recruited for personal exposure monitoring. HAP exposure measurements during traditional stove use occurred during May–July 2016 and after LPG stove distribution in November–December 2016 (Table 1). All participants used a traditional clay stove and readily available biomass fuels. Only one participant cooked with their traditional stove inside their home; the remainder had their traditional stoves in a separate building or outdoors. Four participants also had access to an electric or other type of stove kept inside their home. Cigarette smoking occurred inside 16 of the homes. Locally made, two burner LPG stoves were installed inside the home, not in a separate kitchen outside the home, during the 3 month intervention phase. Each participant received 3 LPG cylinders during the intervention phase. Stoves and cylinders were free to study participants.

The MicroPEM™ (RTI, Research Triangle Park, NC, USA), a low burden and wearable (240 g with batteries) particulate matter (PM) exposure monitor, assessed personal exposure to HAP pre- and post-intervention. Dual stage impactors aerodynamically selected the PM_2.5_ fraction for collection on a pre-weighed 25 mm PTFE filter (Zefon International, Ocala, FL, USA). Gravimetric analysis of the filter determined the average PM_2.5_ personal exposure concentration over the 24 h measurement period [39]. Black carbon (BC) and brown carbon—environmental tobacco smoke (BrC-ETS) mass on the filter were measured by multiwavelength optical transmittance [40]. The MicroPEM measured PM_2.5_ real-time concentrations every 10 s via light scattering nephelometry. Nephelometer data were corrected such that the integrated average was equal to the corresponding gravimetric concentration then consolidated to a 1 min average. Participant compliance with wearing the MicroPEM in the pocket of a sash located on the upper left of the women’s chest was calculated from the accelerometer data [41]. Statistical analysis (SAS Enterprise v7.1, SAS, Cary, NC, USA) validated the MicroPEM filter and nephelometer data. Sample periods less than 22 h and outlier PM_2.5_ concentrations were flagged as invalid. The invalid samples were confirmed by visual analysis of the MicroPEM data file and the filter. PROC GLM compared PM_2.5_ mass and species concentration reductions pre- and post-intervention.

## 3. Results

We collected valid paired baseline and intervention data from 22 of 30 women (73%) and 87% of the total samples (52 of 60). All eight invalid MicroPEM samples occurred during baseline. Mishandling of MicroPEM filters by the data collection team before or after sample collection invalidated six samples. The remaining two invalid samples occurred because the MicroPEM experienced battery failure that resulted in sample collection periods shorter than 22 h. All study participants wore the MicroPEM an average of 77 ± 6% (min–max: 49–93%) and 69 ± 7% (min–max: 43–89%) of their time awake during baseline and intervention, respectively. The wearing compliance values show participants wore the MicroPEM consistently, indicating that the data collected are representative of their exposure.

Mean gravimetric PM_2.5_ concentrations during baseline and intervention were comparable: 81.3 µg/m^3^ vs. 75.3 µg/m^3^ (*p* = 0.518) (Table 2). Black carbon comprised 56.4 µg/m^3^ (69.3%) and 68.7 µg/m^3^ (91.2%) of the PM_2.5_ mass during baseline and intervention, respectively, and were statistically similar (*p* = 0.646). BrC-ETS concentrations were low: 4.1 µg/m^3^ during baseline and less than 1 µg/m^3^ during intervention (*p* < 0.0001).

Figure 1 is a percentile distribution plot of the 1 min average PM_2.5_ nephelometer concentrations across all participants with valid data during the baseline and intervention phases. The nephelometer concentrations during baseline exhibited a log-normal distribution, as evidenced by the linearity of the line, until the inflection point at approximately 25 µg/m^3^ (65th percentile). The change in slope at that point suggests a strong source of PM_2.5_, likely from traditional stoves, was present. Nephelometer concentrations during the LPG intervention were higher than baseline values until the 94th percentile was crossed. The exposure concentration distribution during intervention was log-normal until the 97th percentile, when a slight increase in slope occurred. The small slope change suggests participants had a sporadic but weak secondary source of PM_2.5_ exposure during the intervention phase.

Figure 2 shows the mean baseline and intervention PM_2.5_ nephelometer concentrations for each hour of the day. The average hourly data suggest that PM_2.5_ exposure was lower when cooking with the LPG stove. During peak cooking hours of 06:00 to 08:00, 12:00 to 14:00, and 16:00 to 18:00, the mean PM_2.5_ concentrations during the baseline concentrations exceeded the intervention concentrations (Appendix A). The mean MicroPEM nephelometer concentration during cooking periods with LPG was 84.6 µg/m^3^ vs. 136.5 µg/m^3^ with traditional stoves. The width of the 95% confidence intervals during these periods of cooking were also wider during baseline than during intervention.

However, the background PM_2.5_ concentration during intervention was higher than during the baseline phase. The higher concentrations during the intervention phase were prominent overnight, starting at 19:00 h and continuing to 07:00 h. This finding suggests a source of ambient or HAP PM_2.5_ besides cookstove emissions influenced the participant’s exposure and caused their 24 h average PM_2.5_ exposure during each phase to be similar. The higher PM_2.5_ background concentrations were also evident during the day from 09:00 to 14:00 h, but the differences were not as strong. To elucidate the PM_2.5_ contributions from traditional and LPG fuel use, we estimated the average background PM_2.5_ concentrations and subtracted that value from the hourly averages (Figure 3, Appendix A). The background corrected baseline and intervention PM_2.5_ concentrations during cooking were 103.5 μg/m^3^ and 71.5 μg/m^3^, respectively. Background corrected values during non-cooking periods were 20.8 μg/m^3^ and 21.8 μg/m^3^ for baseline and intervention, respectively. Daily mean concentrations were 53.4 μg/m^3^ and 38.4 μg/m^3^ for baseline and intervention, respectively. Note, the elevated intervention phase PM_2.5_ concentrations during the overnight hours remained after correction for background PM_2.5_.

## 4. Discussion

Using personal, real-time PM_2.5_ exposure data, this feasibility study assessed the effect of an LPG cookstove intervention on HAP reduction in rural Bangladesh. Our findings suggest that the use of LPG fuel reduced peak HAP exposure. Aside from this main finding, the personal level data strongly suggest that higher personal PM concentrations present during the intervention phase of the study were caused by ambient PM sources that were not specifically measured in this study. Additionally, the validity of the MicroPEM data collected showed personal exposure measurements were feasible with proper training and support.

Continuous measurements of personal exposure suggested that the use of cleaner LPG cookstoves reduced peak HAP exposure. The non-significant reduction in exposure as measured by the daily average concentration, on the basis of gravimetric mass or nephelometer measures, could suggest the LPG intervention was not effective at reducing HAP exposure. However, we determined from the time series PM_2.5_ concentrations measured by the nephelometer that the background PM_2.5_ concentration during the intervention phase was higher than during baseline. After subtracting the mean background concentrations for baseline and intervention, it became apparent that PM_2.5_ exposures were lower during cooking periods with LPG fuel than with traditional fuels. The absence of an increase in PM_2.5_ concentration around 12:00 h, lunch, during the intervention phase strongly suggests any PM_2.5_ increase in the morning or evening originated from other sources of HAP. Overall, we estimate that the use of LPG fuel reduced exposure to HAP during cooking periods by approximately 31%, from 103.5 μg/m^3^ to 71.5 μg/m^3^.

This reduction in PM_2.5_ concentrations provided additional evidence for the effectiveness of a widespread LPG intervention, which is a similar result compared to several other studies. A pilot study in India as part of the HAPIN trial reported mean personal PM_2.5_ exposures in homes that used LPG were 36 μg/m^3^ versus 75 μg/m^3^ in homes using biomass fuels [42]. In the GRAPHS study, there was a 32% reduction in PM_2.5_ exposure in the LPG arm compared to the control arm [27]. In rural Nepal, the mean PM_2.5_ concentration was 442 μg/m^3^ with LPG stove usage, compared to a 1380 μg/m^3^ average concentration associated with traditional biomass stoves [29]. Similarly, there was a 33% reduction in personal exposure to PM_2.5_ in pregnant women associated with using LPG for cooking instead of biomass in an effectiveness study in rural Mexico [26]. Another study in Guatemala showed similar statistics, supporting the health benefits of LPG stove usage when compared to more traditional methods [25]. Despite this significant finding that LPG stoves can reduce the level of PM_2.5_ in a household, PM concentrations in our study still exceeded the WHO recommendation for an annual mean of 35 μg/m^3^ and a 24 h mean of 25 μg/m^3^. This suggests other contributing factors to pollution levels other than cookstoves.

Seasonality was an external factor that affected the background PM_2.5_ in the study’s location. Baseline exposure assessment was conducted from May through June and coincided with the monsoon season for Bangladesh. We suspect the daily rainfall reduced the background ambient PM_2.5_ concentrations that would infiltrate into participant’s homes [43]. The LPG intervention and exposure assessment was performed in November and December, dry months that are also periods when brick production and rice mills are more active. The research team counted 33 rice mills and previous research estimated nine brick kilns within our study area (Appendix A) [44]. The PM_2.5_ emissions generated using traditional fuels by these facilities would create an elevated ambient PM_2.5_ concentration that could easily infiltrate into the participant’s homes and be measured by a personal exposure monitor. The homes in the study area are frequently constructed of natural materials that have high infiltration factors [5].

Additionally, other participant behaviors could have contributed to the increase in PM_2.5_ concentrations between 18:00 and 23:00 h during the intervention phase. Bangladesh experiences cooler temperatures in November and December, which increases traditional fuel use to heat the participant’s homes, especially evident since 29 of the 30 participants kept their traditional stoves in a separate kitchen [45]. Our survey data determined stove stacking was common as participants tried to conserve their LPG fuel for cooking and use traditional fuels for other activities such as heating [30]. Anecdotal evidence also suggests study participants used household products that produce significant amounts of PM_2.5_, especially black carbon, during the winter. Mosquitos are a nuisance in November–December, and mosquito coil use is likely high during those months leading to the measured elevated PM_2.5_ concentrations [46,47]. Use of kerosene lamps as a light source is common in rural Bangladesh and was observed by study personnel during the intervention phase [48,49]. Kerosene combustion is known to be a significant source of HAP and black carbon [50,51,52]. These two sources could have contributed the more than more than 90% of the average PM_2.5_ mass during the intervention phase.

Low burden, real-time exposure monitors such as MicroPEM devices are proving, through various studies including this one, to be a major advancement that improves cost-effectiveness, efficiency, and productivity of exposure-health studies [27,53,54,55]. When they were combined with the Beacon method, another recently developed tool, researchers began to obtain a complete picture of the microenvironmental PM_2.5_ concentrations within a household that determine a person’s exposure and associated health [56]. Measuring personal level exposure data with a MicroPEM and a Beacon, instead of using stationary monitors, reduces the exposure misclassification that could weaken epidemiological studies. Stationary monitors are typically placed closer to the stove than the study participants, which can also cause an overestimation of exposure levels [55]. The MicroPEM-Beacon approach provides microenvironmental exposure data that could account for other household or ambient sources of PM_2.5_ when assessing the intervention effectiveness. This approach could be especially useful for non-cooks who spend more time away from the stove and have greater exposure to HAP sources [55].

The high data quality of this feasibility study is comparable to previous studies conducted in low-resource and low-income environments. Our overall MicroPEM sample validity was 83% or 73%, calculated on a matched pair basis for each participant. Chillrud et al. and Chartier et al. had MicroPEM sample validity rates of 80% and 97%, respectively [27,55]. Our data validity rate could have been higher if the sample size was larger because problems handling MicroPEM filters were not identified until the baseline sample collection for the 30 participants was almost complete. High data quality resulting from this study was also largely due to high wearing compliance for the 24 h period among participants and real-time data, which accurately depicts exposure on a daily, personal level and contributes greatly to temporal information. This points to multiple sources of exposure related to this study, helps us to focus on which factors have the greatest influence on HAP levels, and disentangles the influence of a clean cooking intervention from other ambient or HAP sources.

Exposure misclassification was low because our mean wearing compliances were 77 ± 6% and 69 ± 7% for baseline and LPG use phases, respectively. This was 37% greater than the minimum of 40% recommended for personal exposure measurements [41,55,57]. This offers evidence that the MicroPEM is a low-burden, feasible, and effective method of personal exposure monitoring. Wearing compliance is important in determining the overall effectiveness of MicroPEMs because it provides insight into participant engagement and overall study burden; higher engagement and enthusiasm is associated with higher wearing compliance [53,55,57]. Additional factors must be considered to evaluate burden from a well-rounded approach. For example, the burden of more than two consecutive days of personal PM_2.5_ exposure data collection can increase overall burden and reduce wearing compliance [53]. A much higher potential for better compliance is possible, as shown in previous studies. A similar Sri Lankan study that used the MicroPEM yielded a wearing compliance of 87.2% for the first 24 h of data collection [58]. Likewise, a personal PM exposure study of pregnant women conducted in the USA reports a mean compliance of 56%, which supports the minimal burden of MicroPEM devices [59]. Higher validity of data can be assumed from these MicroPEM measurements since periods of non-compliance in these studies most often occurred during non-cooking times.

This study followed a train-the-trainer (TTT) approach, which is an educational public health preparedness model where an organizing institution identifies potential trainers connected to the community that are targeted for training; these trainers are provided with instructional tools and programmatic guidelines with the intention of passing down this education to target audiences [60]. TTT is regarded as a sustainable method due to its long-term sustainability, cost-effectiveness, and ability to maximize social capital and community connectivity [60]. Repetitive training within this model was required to minimize user error. This training model combined with the implementation of monthly data quality review provided us confidence that we could obtain 90% valid sample collection during the full clinical trial.

## 5. Conclusions

This feasibility study found that the use of LPG fuel reduced exposure to PM_2.5_ during cooking. Using the real-time data that it provided, we observed a reduction of PM_2.5_ exposure during peak cooking times. This points to evidence that interventions regarding cooking behavior and technology reform could help drastically reduce pollution levels and therefore improve maternal and neonatal health outcomes. In addition, it is evident that real-time, low burden PM_2.5_ exposure data are a valuable tool to evaluate the effectiveness of a clean cook-stove intervention; this type of exposure analysis consists of each valid participant’s 1 min PM_2.5_ exposure data over the 24 h sampling period. The wearable, cost-effective, and efficient MicroPEM device offers significant promise for the future of personal air pollution exposure research. The real-time monitoring approach employed by this technology was implemented in the clinical trial that is currently underway.

This feasibility study has several strengths that advance exposure research. One important strength is the high wearing compliance, which shows that the participants thought the MicroPEM was sufficiently enough of a low burden to wear most of their time awake. This high compliance produced valid data that are representative of true exposure levels. Additionally, the errors causing invalid data were addressed and resolved among field technicians and handlers, strengthening the full clinical trial before it fully began in September 2019. The real-time monitoring made possible by the MicroPEM device provides a more accurate estimate of exposure levels since the monitors are more personalized to the wearer; for example, non-cooks and other house dwellers who spend less time away from the stove will likely experience less exposure compared to cooks. The more confident real-time data provided through the methods of this study is crucial in reducing personal exposure.

The feasibility study also highlighted improvements to the exposure monitoring protocol for implementation during the current clinical trial. The final clinical trial design incorporated longitudinal PM_2.5_ exposure assessment of the participants in both arms at three times that roughly corresponding to three different seasons. To confirm the positive bias in personal exposure levels from ambient air pollution, we deployed PurpleAir sensors outside our two field offices for continuous measurement of ambient PM_2.5_ throughout the clinical trial. We also added collection of questionnaire and observational data on kerosene lamp use, mosquito coils, biomass fuel use for household heating, and other potential sources of HAP.

This feasibility study emphasizes the need for real-time, personal level data as we move forward with the full clinical trial. The elevated PM baseline we report post-intervention shows that other sources besides cookstove emissions contribute to HAP, hence the need for continuous personal to understand external sources that could impact measured PM_2.5_ exposure concentrations and the LPG cookstove intervention efficacy. However, to achieve the WHO indoor air quality standard, other HAP sources and the infiltration of ambient PM need controlled.

## Figures and Tables

**Figure 1 ijerph-19-00482-f001:**
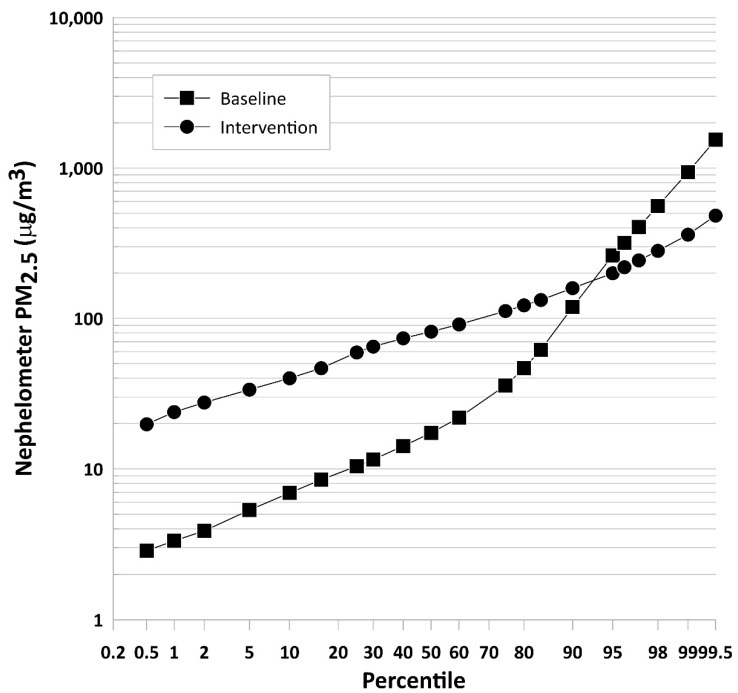
Probability distribution of the valid, 1 min average PM_2.5_ nephelometer concentrations measured during baseline and intervention. The percentiles are the fraction of the exposure data less than the corresponding concentration.

**Figure 2 ijerph-19-00482-f002:**
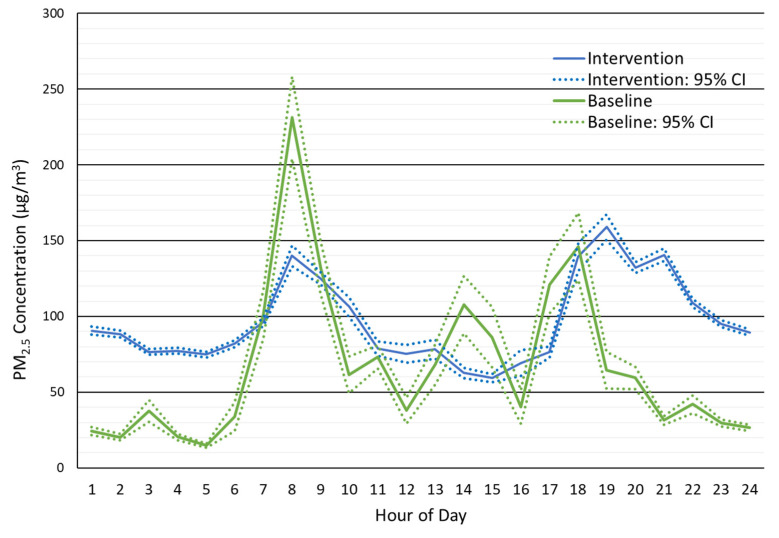
Hourly PM_2.5_ concentrations measured during the baseline and intervention phases. Mean and 95% confidence intervals are shown.

**Figure 3 ijerph-19-00482-f003:**
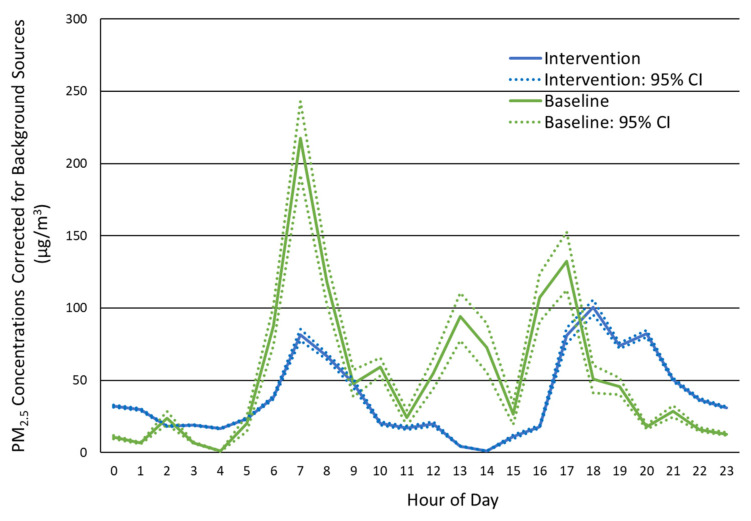
Baseline and intervention phase hourly PM_2.5_ concentrations corrected for background sources to emphasize the contribution of cookstove emissions. Mean and 95% confidence intervals are shown.

**Table 1 ijerph-19-00482-t001:** Cooking, fuel, and tobacco use characteristics.

Characteristic	Response	Number
Primary household cook	Study participant	30
Kitchen location	Separate building used as kitchen	26
(Traditional stove)	Home, separate room from sleeping	1
	Home, same room used for sleeping	0
	Outdoor	3
Primary stove	Traditional, clay stove	30
Stove used other than cooking	No	30
Typical cooking fuels used	Cow dung	28
	Wood, bamboo	28
	Straw, leaves, crop residue	27
	Husks, grass	11
	Kerosene	0
	LPG	0
	Electricity	2
	Other	2
Smoking inside home	Yes	16
	No	14
Number of smokers	1	11
	2	5

**Table 2 ijerph-19-00482-t002:** PM_2.5_, BC, and BrC-ETS concentration distributions measured during the baseline and intervention phases.

	PM_2.5_ (μg/m^3^)		BC (μg/m^3^)		BrC-ETS (μg/m^3^)	
	Baseline	Intervention	Baseline	Intervention	Baseline	Intervention
Mean	81.3	75.3	56.4	68.7	4.1 *	0.2 *
SD	43.8	19.0	20.2	14.8	7.6	0.7
Median	63.1	91.7	54.0	67.2	0	0
IQR	59.0	23.5	24.3	21.0	5.7	0

* *p* < 0.0001.

## Data Availability

Processed and original data are available upon request.

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
