# Peer review of "Pregnant Women’s Exposure to Household Air Pollution in Rural Bangladesh: A Feasibility Study for Poriborton: The CHANge Trial"

_ijerph, 2022, doi:10.3390/ijerph19010482_

Round 1
Reviewer 1 Report
Manuscript ijerph-1500111 “Pregnant Women’s Exposure to Household Air Pollution in Rural Bangladesh: A Feasibility Study for Poriborton: the CHANge Trial” details the PM exposure results obtained using a MicroPEM in a pilot study for an LPG intervention study in Bangladesh. These data were collected in 2016 and provide some interesting insights into PM exposures associated with LPG intervention. These data are unique and important findings that will contribute to the finding of similar studies that have and are being conducted.
General Comments:
- Recommend conducting a bit more thorough literature review to include more recent LPG intervention studies. A few that come to mind that seem to be absent in the referenced literature are the Household Air Pollution Intervention Network (HAPIN) and the LPG adoption in Cameroon Evaluation (LACE).
- Check units throughout. ug/m3 seems to be incorrectly formatted in some but not all instances in the publication.
- Reference and discuss SI material in the main text.
- The manuscript could provide more reflection on the limitations of the study and possibly the full study if it was completed after this feasibility study. Some of the limitations that come to mind are:
- The lack of stationary area monitor(s) in the community to try and account for background PM levels that were elevated in the intervention months.
- The lack of monitoring of the traditional stoves that may have been used for heating or cooking in the intervention months and contributed to elevated PM levels. Either via temperature loggers or stationary PM samplers.
- Plans in the main study to collect baseline and intervention PM data to allow for PM data collection during multiple seasons to try and account for background PM levels.
Specific Comments:
Introduction
- Abstract in the pdf appears to have issues with units as mg/m3 and g/m3 seem to be used whereas, I assume these should be ug/m3.
- Lines 57-59 the abbreviation for what I assume are absolute risk reduction and odds ratio are used but haven’t been defined. Please define.
- Lines 59-61, please specify some of the important risk factors that were not accounted for in previous studies.
- Line 61 mentions low birth weight and then line 66 mentions low birth rate, should line 66 be low birth weight instead of low birth rate? Generally, I find this paragraph (lines 51-69) a bit hard to follow and I recommend reworking this paragraph for clarity.
- Line 82-84 mentions two studies have shown a 33% reduction. Did both studies determine this exact reduction? If not please clarify provide additional information to demonstrate the reductions in each study.
- Line 92. Availability of which fuels are limited? LPG?
Methods
- Table 1 could use some more description in the text. Also it’s unclear why there are only 27 values for “Kitchen Location” when all of the other table items appear to have a total of 30 responses.
- Methods don’t mention if the traditional stoves were destroyed and if not, if there was any monitoring in place to determine if some traditional stove use was still occurring during the LPG intervention.
- Was the LPG provided for free and unlimited use?
- While these last two items are discussed in ref 30 it would be helpful to provide some high level details such as these in this manuscript for clarity.
Results
- Lines 173-176 mention that the intervention peaks are lower during peak cooking hours. While this is clearly shown in the morning, I would disagree that the intervention evening peak is lower. While it is technically true between 1400-1800, there are substantially higher PM values for longer in the evening. This larger and longer peak is also shown in Figure 3 which has background removed. I think the total area under the curves in the evenings or 24 hours should be compared to demonstrate/validate the statements made in lines 190-195. The discussion does mention that some stove stacking may have occurred for heating and other cooking which helps explain this additional peak for the intervention. I would recommend adding a short mention of this here in the results that stove stacking for home heating etc. may have contributed to this evening peak.
- Also, there really isn’t a midday peak for intervention but there is one for baseline which is a bit curious. More discussion on this would be helpful.
- Recommend calculating estimated PM statistics for data shown in Figure 3 that has background PM removed.
- Need to explain in the Methods how the background concentration was estimated and what those values were for both intervention and baseline.
Discussion
- Lines 211, 220, 225 have issues with units. g/m3 instead of ug/m3.
- Lines 210-211 is redundant as already mentioned in the Results. “The mean 24-hour PM2.5 concentrations 210 across all participants are 81.3 g/m3 at baseline and 75.3 g/m3 during the intervention.”
- Lines 219-220. It’s unclear how the 103.5 and 71.5 exposure values were calculated as those don’t appear to be in the results section.
- Lines 224-225 use the avg PM concentrations whereas all other studies you provide a % reduction. Please standardize for consistency.
- There has been at least one study (Jack, DW 2021, http://dx.doi.org/10.1136/bmjgh-2021-005599) that didn’t show health improvements due to the LPG interventions. The PM data presented in that work appears to have similar results to what is shown in this publication. I suggest adding some discussion (near lines 230-233) around this and other similar publications if they exist on how there is a potential that LPG interventions may not improve health outcomes.
- Lines 235-246. Reference the SI map when discussing the rice mills and kilns in lines.
- Lines 252-259. Would more kerosene use be seen in Nov-Dec as a result of slightly shorter days? Also are mosquito coils not typically used year-round if used at all?
- Last sentence starting on line 271 seems out of place or poorly tied to the previous sentence. Please revise with an improved transition within the paragraph.
Conclusions
- Lines 340-341 state that this shows LPG intervention can reduce HAP to WHO standards. However, this is assuming no additional background PM. While there is mention of background PM in the following sentence, I think this statement needs to be reworded to account for the fact that just LPG interventions aren’t likely to improve the HAP levels to WHO levels unless these background and stove stacking for heating (as mentioned in the discussion) are also accounted for. Please edit the conclusions to reflect this.
- The last sentence of the conclusion section doesn’t seem to make sense to me. Please revisit and edit as necessary.
Author Response
Dear Reviewer,
We appreciate your thoughtful comments on our draft manuscript. We edited or added to the manuscript to address all but one of your comments. As further explained below, our study design was not sufficient to correlate LPG use with improved health.
General Comments:
- Recommend conducting a bit more thorough literature review to include more recent LPG intervention studies. A few that come to mind that seem to be absent in the referenced literature are the Household Air Pollution Intervention Network (HAPIN) and the LPG adoption in Cameroon Evaluation (LACE).
- Response: Thank you for the suggestion. Two papers from the HAPIN study and one from the LACE study were added that support the rationale and results of this paper.
- Check units throughout. ug/m3 seems to be incorrectly formatted in some but not all instances in the publication.
- Response: Fixed. Thank you for identifying the error. Note, we think the journal created these errors when formatting the draft manuscript.
- Reference and discuss SI material in the main text.
- Response: Done.
- The manuscript could provide more reflection on the limitations of the study and possibly the full study if it was completed after this feasibility study. Some of the limitations that come to mind are:
- The lack of stationary area monitor(s) in the community to try and account for background PM levels that were elevated in the intervention months.
- The lack of monitoring of the traditional stoves that may have been used for heating or cooking in the intervention months and contributed to elevated PM levels. Either via temperature loggers or stationary PM samplers.
- Plans in the main study to collect baseline and intervention PM data to allow for PM data collection during multiple seasons to try and account for background PM levels.
Response: Excellent comment. We added a discussion of study limitations to the Conclusions. We also described changes to the study protocol implemented for the clinical trial to address these limitations.
Specific Comments:
Introduction
- Abstract in the pdf appears to have issues with units as mg/m3 and g/m3 seem to be used whereas, I assume these should be ug/m3.
- Response: Fixed
- Lines 57-59 the abbreviation for what I assume are absolute risk reduction and odds ratio are used but haven’t been defined. Please define.
- Response: Fixed
- Lines 59-61, please specify some of the important risk factors that were not accounted for in previous studies.
- Response: Added
- Line 61 mentions low birth weight and then line 66 mentions low birth rate, should line 66 be low birth weight instead of low birth rate? Generally, I find this paragraph (lines 51-69) a bit hard to follow and I recommend reworking this paragraph for clarity.
- Response: Changed “rate” to “weight.” Also split the paragraph into two paragraphs to improve clarity.
- Line 82-84 mentions two studies have shown a 33% reduction. Did both studies determine this exact reduction? If not please clarify provide additional information to demonstrate the reductions in each study.
- Response: Yes, both studies found 33% reduction when rounded to the nearest integer.
- Line 92. Availability of which fuels are limited? LPG?
- Response: Clarified – LPG.
Methods
- Table 1 could use some more description in the text. Also it’s unclear why there are only 27 values for “Kitchen Location” when all of the other table items appear to have a total of 30 responses.
- Response: Discussion of Table 1 was added, and text split into two paragraphs. The missing 3 kitchens were outdoors.
- Methods don’t mention if the traditional stoves were destroyed and if not, if there was any monitoring in place to determine if some traditional stove use was still occurring during the LPG intervention.
- Response: The participant’s traditional stoves were not destroyed. However, all but one participant kept their traditional stove in a communal kitchen that was not attached to their house. Their LPG stove and fuel were kept inside their home.
- Was the LPG provided for free and unlimited use?
- Response: Text clarified. Stoves and LPG were free. Three LPG cylinders were provided over 3 months.
- While these last two items are discussed in ref 30 it would be helpful to provide some high-level details such as these in this manuscript for clarity.
- Response: Text added to convey relevant information for interpretation of the results presented in this manuscript.
Results
- Lines 173-176 mention that the intervention peaks are lower during peak cooking hours. While this is clearly shown in the morning, I would disagree that the intervention evening peak is lower. While it is technically true between 1400-1800, there are substantially higher PM values for longer in the evening. This larger and longer peak is also shown in Figure 3 which has background removed. I think the total area under the curves in the evenings or 24 hours should be compared to demonstrate/validate the statements made in lines 190-195. The discussion does mention that some stove stacking may have occurred for heating and other cooking which helps explain this additional peak for the intervention. I would recommend adding a short mention of this here in the results that stove stacking for home heating etc. may have contributed to this evening peak.
- Response: The intervention peak is lower during the evening cooking times (1600-1800), but the difference was not as pronounced as the breakfast and lunch cooking times. References to Tables S1 and S2 were added to guide the reader to additional details that support the results. Line 185 of original draft was revised to clarify that “another source of ambient or HAP PM2.5 besides cookstove emission influenced the participant’s exposure…” The
- Also, there really isn’t a midday peak for intervention but there is one for baseline which is a bit curious. More discussion on this would be helpful.
- Response: Added this sentence to the Discussion. “The absence of an increase in PM2.5 concentration around 1200 hours, lunch, during the intervention phase strongly suggests any PM2.5 increase in the morning or evening originated from other sources of HAP.”
- Recommend calculating estimated PM statistics for data shown in Figure 3 that has background PM removed.
- Response: Added to Table S2.
- Need to explain in the Methods how the background concentration was estimated and what those values were for both intervention and baseline.
- Response: Added to the text surrounding Table S2.
Discussion
- Lines 211, 220, 225 have issues with units. g/m3 instead of ug/m3.
- Response: Addressed earlier.
- Lines 210-211 is redundant as already mentioned in the Results. “The mean 24-hour PM2.5 concentrations 210 across all participants are 81.3 g/m3 at baseline and 75.3 g/m3 during the intervention.”
- Response: Removed.
- Lines 219-220. It’s unclear how the 103.5 and 71.5 exposure values were calculated as those don’t appear to be in the results section.
- Response: Added the average cooking exposure PM2.5 concentrations to the results.
- Lines 224-225 use the avg PM concentrations whereas all other studies you provide a % reduction. Please standardize for consistency.
- Response: Our data was updated to provide both PM concentrations and % reduction throughout. This approach eases the comparison with previous studies that provided either PM concentrations or % reduction (and sometimes both).
- There has been at least one study (Jack, DW 2021, http://dx.doi.org/10.1136/bmjgh-2021-005599) that didn’t show health improvements due to the LPG interventions. The PM data presented in that work appears to have similar results to what is shown in this publication. I suggest adding some discussion (near lines 230-233) around this and other similar publications if they exist on how there is a potential that LPG interventions may not improve health outcomes.
- Response: We are aware of the findings of Jack et al. The sample size and overall design for our study reported here was not sufficient to associate LPG intervention with health outcomes. From what we learned in this feasibility study, we hope our protocol will provide the data to support or not support a correlation between LPG intervention and improved perinatal and neonatal health outcomes. Either way, we will make sure to discuss your point in detail when we report the clinical trial findings.
- Lines 235-246. Reference the SI map when discussing the rice mills and kilns in lines.
- Response: Added the reference.
- Lines 252-259. Would more kerosene use be seen in Nov-Dec as a result of slightly shorter days? Also are mosquito coils not typically used year-round if used at all?
- Response: We did not specifically track kerosene and mosquito coil use during this feasibility study. The text was phrased to reflect the anecdotal observations of study personnel and their knowledge of life in rural Bangladesh. Note, the sentence about kerosene lamp use was updated to reflect that study personnel observed lamp use
- Last sentence starting on line 271 seems out of place or poorly tied to the previous sentence. Please revise with an improved transition within the paragraph.
- Response: The last sentence moved to earlier in the paragraph to improve the transition.
Conclusions
- Lines 340-341 state that this shows LPG intervention can reduce HAP to WHO standards. However, this is assuming no additional background PM. While there is mention of background PM in the following sentence, I think this statement needs to be reworded to account for the fact that just LPG interventions aren’t likely to improve the HAP levels to WHO levels unless these background and stove stacking for heating (as mentioned in the discussion) are also accounted for. Please edit the conclusions to reflect this.
- Response: The last paragraph was re-written to address this comment.
- The last sentence of the conclusion section doesn’t seem to make sense to me. Please revisit and edit as necessary.
- Response: The last sentence was fixed when addressing the previous comment.
Reviewer 2 Report
The article examines the exposure of pregnant women to household air pollution in rural Bangladesh and concludes that the use of LPG fuel reduces exposure to PM2.5 during cooking. I think it is a novel perspective.
I would like to make a few suggestions below.
- The details of the tests are described in the abstract in too much detail, but it does not allow the reader to understand the results of the author's article experiments from the abstract reading, and does not provide sufficient effective information in the early stages of reading. It is recommended to reduce the description of the operational details of specific experiments and data in the abstract, mainly reflecting the final results of the experiments and the interpretation of the data and the results or conclusions of the paper.
- Before the "2. Materials and Methods" section was written, most of the descriptions were of the background of the study, and there were not enough references to the research object (rural pregnant women) and the applicability of the research object to the paper. It is recommended that the relevance of the article to the subject of the study should be enhanced and the relevance and applicability of the study to the specific subject of the study should be highlighted.
- The article lacks sufficient literature review support, and the summary of the existing research status in the field is inadequate, and the discussion of the need for research is slightly inadequate. It is suggested that a review of the relevant literature be organized to show the innovation of the article.
- The article selects 30 women for personal exposure monitoring, but the selection criteria for the subjects are not specifically explained, for example, pregnant women in different stages of pregnancy or differences in personal physical quality may also cause differences in the experimental results, so it is difficult to repeat the experiment and get the same results. It is recommended to describe the basis and actual situation of the selection of the research subjects to ensure the rigor of the research experiment.
- The article lacks an obvious scientific question. It is suggested that the valuable scientific question condensed in the article be clearly presented or further highlighted in the article to fully improve the academic research value of the article.
- The article lacks an outlook on its own conclusions. It is suggested that a reasonable outlook on existing research in the field be added to the "5. Conclusions" section, such as future research scholars can do comparative experiments on rural pregnant women at different stages of pregnancy, or further explore the correlation between factors affecting household air pollution and factors in the future.
I hope that the reviewers' suggestions will be helpful to the authors.
Author Response
Dear Reviewer,
We appreciate your thoughtful comments on our draft manuscript. We edited or added to the manuscript to address your comments. How we addressed your comments is described below.
1. The details of the tests are described in the abstract in too much detail, but it does not allow the reader to understand the results of the author's article experiments from the abstract reading, and does not provide sufficient effective information in the early stages of reading. It is recommended to reduce the description of the operational details of specific experiments and data in the abstract, mainly reflecting the final results of the experiments and the interpretation of the data and the results or conclusions of the paper.
Response: Good comment. We edited the abstract to focus on the primary conclusion of the paper. We kept the abstract within the word limit by removing experimental details as you suggested.
2. Before the "2. Materials and Methods" section was written, most of the descriptions were of the background of the study, and there were not enough references to the research object (rural pregnant women) and the applicability of the research object to the paper. It is recommended that the relevance of the article to the subject of the study should be enhanced and the relevance and applicability of the study to the specific subject of the study should be highlighted.
Response: We added text and citations to clarify the study focus was rural, pregnant women’s HAP exposure from traditional stove use to improve perinatal and neonatal outcomes.
3. The article lacks sufficient literature review support, and the summary of the existing research status in the field is inadequate, and the discussion of the need for research is slightly inadequate. It is suggested that a review of the relevant literature be organized to show the innovation of the article.
Response: The HAP, clean cooking fuel intervention, and health literature is extensive. We had to bound our literature review within perinatal and neonatal mortality (overall purpose of CHANge), LPG cooking interventions (intervention implemented), and personal exposure measurements (our evaluation of intervention success). We originally cited 29 articles in the Introduction and another 16 in the Discussion. The other reviewer recommended addition of another 3 articles from the HAPIN study and the LACE study. Although we may have missed a few other relevant articles, we believe the literature review is sufficient and properly organized.
4. The article selects 30 women for personal exposure monitoring, but the selection criteria for the subjects are not specifically explained, for example, pregnant women in different stages of pregnancy or differences in personal physical quality may also cause differences in the experimental results, so it is difficult to repeat the experiment and get the same results. It is recommended to describe the basis and actual situation of the selection of the research subjects to ensure the rigor of the research experiment.
Response: The most relevant inclusions criteria (gestational age and not using LPG for cooking) were added. The reader is referred to Citation 30 for additional details.
5. The article lacks an obvious scientific question. It is suggested that the valuable scientific question condensed in the article be clearly presented or further highlighted in the article to fully improve the academic research value of the article.
Response: The scientific question (three objectives of this paper) is presented in the last sentence of the Introduction.
6. The article lacks an outlook on its own conclusions. It is suggested that a reasonable outlook on existing research in the field be added to the "5. Conclusions" section, such as future research scholars can do comparative experiments on rural pregnant women at different stages of pregnancy, or further explore the correlation between factors affecting household air pollution and factors in the future.
Response: Your comment was similar to a comment from the other reviewer. We expanded our discussion on the strengths and weaknesses identified in this paper, and suggested improvements for future research.